# Structural and thermal analyses in semiconducting and metallic zigzag single-walled carbon nanotubes using molecular dynamics simulations

Ama tul Zahra[1], Aamir Shahzad[1]*, Alina Manzoor[1], Jamoliddin Razzokov[2,3,4], Qurat ul Ain Asif[1], Kun Luo[5], Guogang Ren[6]*

1 Modeling and Simulation Laboratory, Department of Physics, Government College University Faisalabad, Faisalabad, Punjab, Pakistan, 2 Institute of Fundamental and Applied Research, National Research University TIIAME, Tashkent, Uzbekistan, 3 College of Engineering, Central Asian University, Tashkent, Uzbekistan, 4 Department of Biomedical Engineering, Tashkent State, Technical University, Tashkent, Uzbekistan, 5 School of Materials Science and Engineering, Changzhou University, Changzhou, P R China, 6 School of Physics, Engineering and Computer Sciences, University of Hertfordshire, Hatfield, United Kingdom

* aamir.awan@gcuf.edu.pk (AS); g.g.ren@herts.ac.uk (GR)

**Data Availability Statement:** All relevant data are included within the manuscript and Supporting Information files.

## Abstract

Equilibrium molecular dynamics (EMD) simulations have been performed to investigate the structural analysis and thermal conductivity ($\lambda$) of semiconducting (8,0) and metallic (12,0) zigzag single-walled carbon nanotubes (SWCNTs) for varying $\pm\gamma$(%) strains. For the first time, the present outcomes provide valuable insights into the relationship between the structural properties of zigzag SWCNTs and corresponding thermal behavior, which is essential for the development of high-performance nanocomposites. The radial distribution function (RDF) has been employed to assess the buckling and deformation understandings of the (8,0) and (12,0) SWCNTs for a wide range of temperature $T$(K) and varying $\pm\gamma$(%) strains. The visualization of SWCNTs shows that the earlier buckling and deformation processes are observed for semiconducting SWCNTs as compared to metallic SWCNTs for high $T$(K) and it also evident through an abrupt increase in RDF peaks. The RDF and visualization analyses demonstrate that the (8,0) SWCNTs can more tunable under compressive than tensile strains, however, the (12,0) zigzag SWCNTs indicate an opposite trend and may tolerate more tensile than compressive strains. Investigations show that the tunable domain of $\pm\gamma$(%) strains decreases from (-10%$\leq \gamma \leq$+19%) to (-5%$\leq \gamma \leq$+10%) for (8,0) SWCNTs and the buckling process shifts to lower $\pm\gamma$(%) for (12,0) SWCNTs with increasing $T$(K). For intermediate-high $T$(K), the $\lambda$($T$) of (12,0) SWCNTs is high but the (8,0) SWCNTs show certainly high $\lambda$($T$) for low $T$(K). The present $\lambda$($T$, $\pm\gamma$) data are in reasonable agreement with parts of previous NEMD, GK-HNEMD data and experimental investigations with simulation results generally under predicting the $\lambda$($T$, $\pm\gamma$) by the $\sim$1% to $\sim$20%, regardless of the $\pm\gamma$(%) strains, depending on $T$(K). Our simulation data significantly expand the strain range to -10% $\leq \gamma \leq$ +19% for both zigzag SWCNTs, depending on temperature $T$(K). This extension

**Funding:** Dr. Kun Luo was supported by National Natural Science Foundation of China, grant numbers 51874051 and 52111530139. Dr. Guogang Ren was supported by the UK Royal Society, grant number IEC\NSFC\201155, 2021-23. The funders had the following roles in the development of the paper: - National Natural Science Foundation of China: Formal Analysis, Funding Acquisition, Validation and Writing – Review & Editing - UK Royal Society: Formal Analysis, Funding Acquisition, Methodology and Writing – Review & Editing

**Competing interests:** NO authors have competing interests

of the range aims to establish a tunable regime and delve into the intrinsic characteristics of zigzag SWCNTs, building upon previous work.

## 1. Introduction

Investigating the structural properties of nanomaterials is a crucial aspect of nanotechnology having a wide range of potential applications [1]. Over the past two decades, significant progress has been made in this field through the utilization of experimental, theoretical, and computational methodologies. Such investigations have also been performed in the context of nanotubes aiming to enhance their strength, due to their unique mechanical, thermal, and electrical properties as well as their potential applications across diverse fields. Reinforcement techniques are employed to enhance the strength of structural components while maintaining their dimensions by applying molecular dynamics (MD) simulations [2]. A methodology employed for investigating the structural and mechanical characteristics of materials involves the incorporation of carbon nanotubes (CNTs), with a particular focus on single walled CNTs (SWCNTs). Amidst the diverse classifications of CNTs, the zigzag-oriented SWCNTs have been gained notable interest due to their potential applications in nanoelectronics, nanomechanics, and thermal management [3]. The thermal and electronic properties of a zigzag CNTs with a given $(n, m)$ chirality can be determined by the crystallographic unit cell vector formula $n—m = 3i$ or $2n + m = 3i$ (where $i$ is an integer and $n \geq m$). Zigzag CNTs that meet either of these criteria demonstrate metallic characteristics, whereas those do not fulfill either criterion display semiconductor behavior [4, 5]. Hence, the study involves an investigation into the behaviors of both metallic and semiconducting zigzag SWCNTs under compressive and tensile strains. Furthermore, it aims to characterize the structural and mechanical properties of these nanotubes by employing the radial distribution function (RDF). This analysis is a valuable contribution to the field of nanomaterials research which can lead to the development of high-performance nanocomposites for various applications, such as in the field of energy storage, electronics, sensors and nanocomposites [6]. Moreover, the mechanical properties and behavior of SWCNTs, including their stiffness, strength, and ductility, hold the promise for utilization in the creation of high-performance structural materials [7].

The present paper focuses on the analysis of structural properties of two types of zigzag SWCNTs with corresponding thermal conductivity for a wide range of temperatures ($T$) and strains $\gamma(\%)$. Many studies have attempted to gain a comprehensive understanding of the thermal characteristics of CNTs through the investigating of their structural behavior. These studies have focused on a range of parameters, including nanotube length, diameter, temperature, radius, grain boundaries, and stress/ strain etc., using MD simulations [8–13]. MD simulation is a promising tool for mechanical properties prediction to show a correlation between experimental and simulated data [14]. The conclusion drawn is to fine-tune mechanical performance of nanocomposites for various applications. Goel *et al.*, [15] focused on how vacancy and topological defects of armchair and zigzag SWCNTs disturb their properties when subjected to buckling conditions and tensile loadings. Working on the mechanical properties of carbon and boron nitride nanotubes using MD simulation, Vijayaraghavan and Zhang [16] identified a threshold radius above which elastic modulus remains constant. In the design of nanostructures and thermal management of nano devices, it is crucial to have a complete understanding of the effects of strain on their structural and a thermal property is pivotal. This knowledge is essential due to the frequent exposure of these materials to tensile or compressive strains in

real-world applications. Yakobson *et al.*, [17] utilized a many-body interatomic potential to subject SWCNTs and double-walled CNTs to significant strains and examined their behaviors at different temperature. The results showed that the thermal conductivity of CNTs decreases with increasing temperature under large strains, highlighting the need for careful consideration of strain effects in the design and optimization of CNT-based nanocomposites for thermal applications. Jian Li *et al.*, [18] used MD simulations to investigate the mechanical and thermal properties of a CNT@GNT hybrid nanotube. The findings demonstrated a dependence of the mechanical properties on the nanotube chirality of its components, and indicated that strain engineering can effectively modulate thermal conductivity. Phonon analysis reveals that the strain effect arises from alterations in phonon group velocity and scattering. The thermal properties of materials play a critical role in determining their suitability for various applications, such as in aerospace, automotive, and biomedical engineering [19]. The unique properties of CNTs, including their high strength and stiffness, position them as promising candidates for the development of cutting-edge materials with enhanced mechanical characteristics [20]. For example, the length and diameter of CNTs can be tuned to optimize their sensitivity and selectivity for a specific analyte. In addition, the mechanical properties of CNTs can be utilized to design nano mechanical sensors capable of detecting changes in mass or force [21]. One potential application of the findings on the mechanical properties of strained nanotubes could be in the development of high-performance nanoelectromechanical systems, which could have applications in sensing, actuation, and communication technologies. Furthermore, these finding could also find application in the design and fabrication of high-strength and lightweight materials tailored for structural and aerospace applications. Additionally, the exploration of the electronic properties in strained nanotubes could pave the way for innovations in electronic device design including the creation of novel devices like field-effect transistors (FETs) and other types of sensors [22].

The primary objective of this study is to examine the structural and thermal properties of semiconducting and metallic zigzag SWCNTs using EMD simulations, both with and without the application of strains. For both cases of zigzag SWCNTs, we have employed the RDF analysis and the Green-Kubo relation (GKR) for the thermal conductivity calculations. These methods allow us to investigate the structural behaviors and thermal conductivity across a broad temperature range $T$(K) and varying strains $\pm\gamma$(%). This study aims to provide insights linkage between structural changes and corresponding thermal conductivity of the CNTs. Additionally, this research can offer significant insights into the use of RDF analysis for the comprehensive structural characterization of nanostructures, which has implications for the development of other nanocomposites and nanomaterials in the future.

## 2. Model and computational technique

This section provides the details of computational scheme that are implemented for zigzag SWCNTs with and without strains and varying temperature. MD simulation is an indispensable tool for investigating structural and thermal properties of zigzag SWCNTs. In the present work, the focus is on investigating the structural properties of zigzag SWCNTs, thermal conductivity and how the behavior of these CNTs is affected by changes in $T$(K) and $\pm\gamma$(%) values. We have chosen a zigzag SWCNTs (8,0) with radius $r = 3.820$Å while zigzag SWCNTs (12,0) with radius $r = 5.730$Å for the simulation box and are shown in Fig 1. Both zigzag SWCNTs are created by using Atomsk computer software (open source code) which is a command line tool for manipulating and converting atomic simulation data [10].

The reactive bond-order (REBO) potential has been employed for the simulation of (8,0) and (12,0) SWCNTs which incorporates a combination of covalent and non-covalent

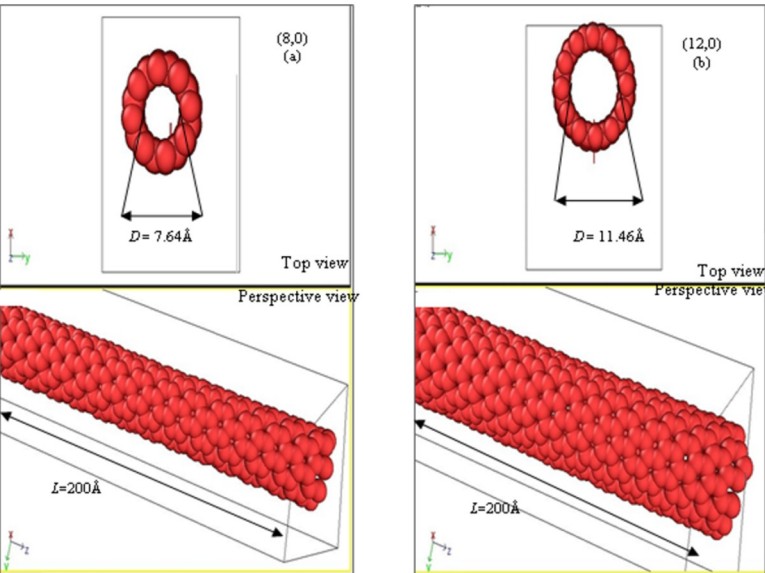

**Fig 1.** Visualization (Top and Perspective views) of (a) semiconducting (8,0) and (b) metallic (12,0) zigzag SWCNTs with diameter of $D$ = 7.64Å and 11.46 Á and length $L$ = 200 Á, respectively.

interactions between atoms and it makes more suitable for simulating chemical reactions and bond breaking in CNTs [23]. The present EMD simulation provides a frictionless atomic system along with applied periodic boundary conditions (PBCs) with their periodic images to model an infinite system. The model can be written in the following form:

$$E_{ij}^{REBO} = f\left(r_{ij}\right)\left(V_{ij}^{R} + b_{ij}V_{ij}^{A}\right), \tag{1}$$

here in the equation, $r_{ij}$ represents the distance between atoms $i$ and $j$, the scalar quantity $b_{ij}$ represents the bond order and adjusts the bonding strength based on the local bonding surroundings. The terms $V_{ij}^{R}$ and $V_{ij}^{A}$ are the repulsive and attractive potential function between the atoms and the cut-off function is denoted by $f(r_{ij})$ that used to restrict interatomic interactions to only the nearest neighbors [24].

Structural and thermal properties of both zigzag SWCNTs having length $L$ = 200Å are studied using EMD simulations. In the present case, a computer software LAMMPS (Large Atomic Molecular Massively Parallel Simulator) is used for all simulations. This software package has been used and designed to simulate a wide range of systems, including soft and condensed matter, biomolecules, and coarse-grained systems. LAMMPS is commonly used in the scientific community for simulating the behavior of materials at the atomic level [25]. The EMD simulations technique consists of solving a set of equations (differential) computed through Newton's end law of motion for all atoms of the zigzag SWCNTs system. In order to measure the continuous atomic trajectories, we have employed the velocity Verlet scheme with the simulation time step of $dt$ = 0.001ps and it allows for long times to obtain the accurate simulation outcomes. The force applying on the $i$th atom $F^{i}$ = (-$\partial E_{ij}$ /$\partial r_{i}$) is computationally obtained by the REBO potential given in Eq (1) between atom $i$ (at $r^{i}$) and atom $j$ (at $r_{j}$) and its periodic images. In practice, first all EMD simulations of both zigzag SWCNT systems run in the isothermal-isobaric ensemble (*NPT*) and then the canonical ensemble (*NVT*) is applied for about 2ns with PBCs in all three Cartesian directions. To ensure there were no temperature gradients, temperature profile is continuously monitored throughout the whole simulation process.

In order to analyze the structural characteristics of zigzag SWCNTs, a basic statistical mechanics approach has been employed, utilizing the radial distribution function or pair correlation function $g(r)$. The RDF is used to study the structural properties of a system, such as the distribution of distances between atoms, the coordination number (i.e., the number of atoms in the immediate vicinity of a given atom), and the degree of ordering or disorder in the system. The general expression of RDF gives the sum over atom pairs and it is represented as.

$$g(r) = \frac{2V}{N_m^2} < \sum_{i<j} \delta(r - r_{ij}) > \cdot \tag{2}$$

The RDF function describes the local grouping around a certain atom and relates to the probability of finding an atom at a distance $r$ from a certain atom. RDF plots are generated using visual MD (VMD) to analyze the structural properties of (8,0) and (12,0) SWCNTs under different $\pm\gamma(\%)$ and $T$(K). To investigate the effect of strains on the structural and thermal properties, the SWCNTs are subjected to compressive $-\gamma$ = (1% to 10%) and tensile $+\gamma$ = (1% to 19%) strains depending on the deformation limit which is different for both strain cases. System temperature is varied from $T$ = 100K to 1000K in order to visualize the complete picture of thermal and structural behaviors under varying strains. Visualizations of the buckling and deformation of SWCNTs under varying strains are analyzed using the OVITO computer software package.

A well known GKR for the thermal conductivity coefficients of particles (uncharged) is given in Ref [26] and this usual GKR of the simple atom has been used to compute the thermal conductivity of CNTs [27] and it is expressed as:

$$\lambda = \frac{1}{3Vk_\mathrm{B}T^2} \int_0^\infty \langle \mathbf{J}(t).\mathbf{J}(0) \rangle dt, \tag{3}$$

here $V$ signifies the system volume, $k_\beta$ for Boltzman constant, $T$ represents the temperature, angular bracket $<—>$ shows the ensemble average over all atoms and it also provides the heat current autocorrelation function (HCACF) [28, 29]. The integral of the time correlation function of the heat current yields the thermal conductivity ($\lambda$) of the zigzag SWCNTs [30]. Heat current is computed by using equation

$$\mathbf{J}(t) = \frac{1}{2V} \sum_i^N \sum_{j=1}^N \mathbf{r}_{ij}.(\mathbf{F}_{ij}.\mathbf{v}_{ij}), \tag{4}$$

where $\mathbf{r}_{ij}$ and $\mathbf{F}_{ij}$ represents the vector distance and force between $i$ and $j$ atoms while $\mathbf{v}_i$ is the velocity of $i^{\mathrm{th}}$ atom [12].

## 3. Results and discussion

For equilibration and convergence of our CNT systems, we have performed two tests for both types of the zigzag SWCNTs through the EMD simulations. Four panels in Fig 2 display the variation of pressure $P(t)$ and temperature $T(t)$ w. r. t time $t$(ps) during the equilibration process of the (8,0) and (12,0) SWCNTs, respectively, performed at room temperature $T$ (= 300K). The $P(t)$ of both types of zigzag SWCNTs fluctuates initially and then gradually converges to a stable value as shown in panels of Fig 2A and 2C. It is observed that the $P(t)$ for the (12,0) SWCNTs converges more sharply with less fluctuations because of its metallic behavior as compared to the (8,0) SWCNTs which acts as the semiconducting behavior of zigzag SWCNTs. This earlier convergence may be due to the metallic behavior and larger diameter of

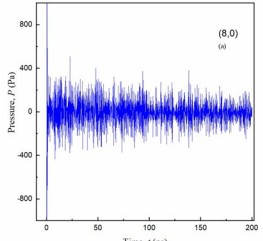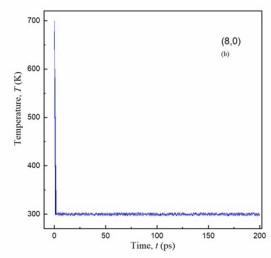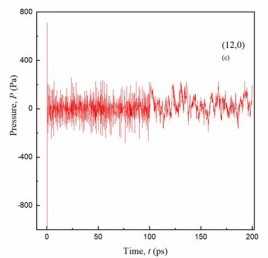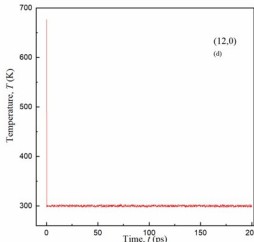

**Fig 2. Convergence response of pressure $P$(Pa) and system temperature $T$(ps) at $T$ = 300K.** The results obtained using EMD simulation method for (a), (b) semiconducting (8,0) and (c), (d) metallic (12,0) zigzag SWCNTs towards equilibrium are shown, respectively.

(12,0) SWCNTs leading to fast equilibration and convergence of the system. Two panels of Fig 2B and 2D represent the variation of $T(t)$ for the same sets [(8,0) and (12,0)] of zigzag SWCNTs. It can be seen from these two panels that the $T(t)$ of both systems initially fluctuates rapidly but both systems eventually soon converge to a stable value throughout the simulation time ($t$ = 200ps). It is noted that the (8,0) nanotube reaches its equilibrium position at a lower system temperature more slowly because of its narrow diameter as compared to the (12,0) nanotube. Overall, these equilibration graphs demonstrate that both zigzag SWCNTs eventually reach a stable equilibrium state after a short period of time but the rate of convergence and the equilibrium $T(t)$ and $P(t)$ depend on the chirality (metallic semiconducting behavior).

The convergence and accuracy of our results were also assessed by plotting HCACF ($t$) graphs for (8,0) and (12,0) SWCNTs with and without strains. These graphs are generated at temperatures of $T$ = 100K, 500K, and 900K, as depicted in Fig 3. Observations reveal that the influence of temperature on the HCACF ($t$) graphs is not significantly pronounced for (8,0) SWCNTs, as can be seen in Fig 3A, 3C and 3E. The fluctuations remain relatively constant throughout the simulation time ($t$ = 100ps) for all $T$(K) range. However, in the case of metallic zigzag SWCNTs, the fluctuations decrease as the $T$ increases and it indicates the faster thermal stability of metallic zigzag SWCNTs (Fig 3B, 3D and 3F). It is examined that the fluctuations in HCACF ($t$) for (8,0) SWCNTs are comparatively higher as compared to fluctuations observed for (12,0) SWCNTs. It may suggest that the metallic zigzag SWCNTs are more resistant to changes and exhibit greater stability when exposed to elevated temperatures.

### 3.1 Effects of strain and structural analysis

In this subsection, we discuss the outcomes of a computational exploration conducted on zigzag SWCNTs. The RDF analysis is employed and focused on assessing the structural stability of strained $\pm\gamma$(%) and unstrained SWCNTs at $100\leq T$(K) $\leq 1000$. The RDF patterns for (8,0) and (12,0) SWCNTs for a range of compressive and tensile strains are shown in four panels of Figs 4 and 5, respectively, at different $T$(K). These panels display the density distribution at $T$ = 100K, 400K, 700K, 1000K (see Fig 4) and $T$ = 100K, 300K, 500K, 900K (see Fig 5), respectively, while the colored lines in the graphs represent varying $\pm\gamma$(%) strains applied to the CNTs. The inset Figs show the heights of first and second peaks of the RDF. Firstly, it is observed that the height of peaks decreases with increasing the system $T$(K). This is because the increased thermal energy causes the atoms vibrate more, which leads to a wider distribution of interatomic distances. As a result, the peak heights decrease and broaden as we move from $T$ = 100K to 1000K [31]. Secondly, the deformation of the CNTs that may due to an applied strain can also be observed in the RDF plots. The change in distance between the carbon atoms is due to an applied strain affects and it results a shift in peak positions and changes

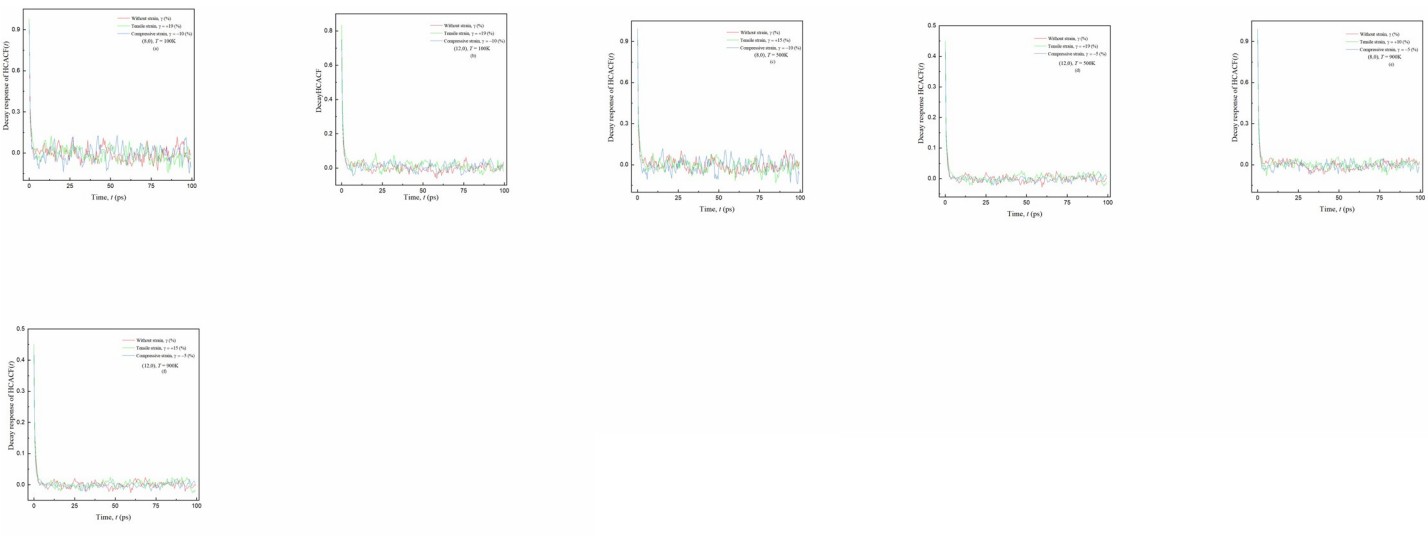

**Fig 3.** Decay response of heat current autocorrelation function HCACF ($t$) for the zigzag SWCNTs at $T$ = 100K, 500K and 900K, respectively, of the (a), (c), (d) semiconducting (8,0) zigzag SWCNTs and (b), (d), (f) metallic (12,0) zigzag SWCNTs. The decay responses obtained through EMD autocorrelation (red color) without strain $\gamma$(%), tensile strain +$\gamma$(%) and (blue color) compressive strain -$\gamma$(%) are shown.

in peak heights. The significance of these changes can be observed by analyzing thermal properties of the nanotube under different $T$(K) and ±$\gamma$(%) [19].

It is examined from four panels of Fig 4 that drop in 1$^{st}$ and 2$^{nd}$ peak heights is large by applying successive compressive strains from -$\gamma$ = 1% to 10% and the difference among the peaks becomes smaller with an increase in $T$(K). The difference in peak heights is pronounced at low $T$ (= 100K) as compared to difference in peak heights at $T$ (= 1000K) for–$\gamma$ strains. We may advance one possible reason in the reduction of peak heights that is due to the compression of the lattice structure of the SWCNTs. In Fig 4A, as the -$\gamma$ strain increases at $T$ = 100K, the atomic separation $r$(Å) decreases and the coordination number of nearest neighbor atoms increases and it leads to a high peak. However, the difference in peaks is less pronounced and reduction of peak height is observed at high $T$ (= 1000K), as shown in Fig 4D. As the thermal vibrations of atoms increase at high $T$(K) and it may cause the average bond length to increase [19] and the effect of compression is relatively smaller.

It can be seen from the first panel of Fig 4A that the (8,0) SWCNT is more stable and can withstand for a wide range of tensile strains +$\gamma$ (= 1% to 19%) before breaking at low $T$ =

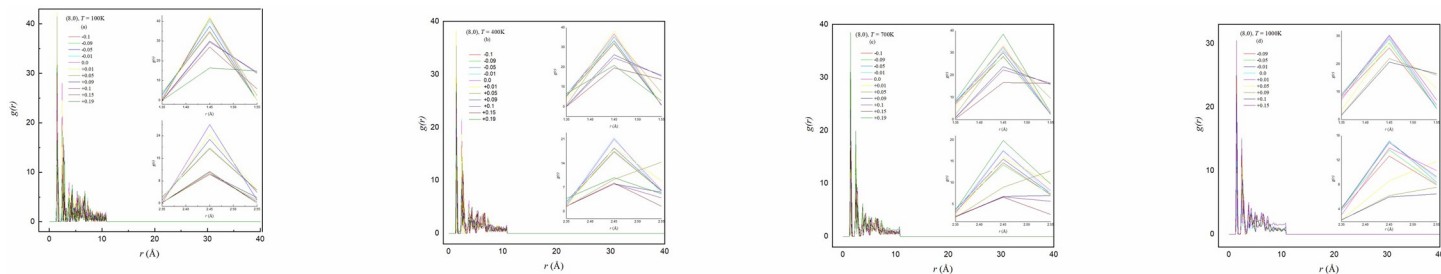

**Fig 4.** Structural analysis test of RDF $g(r)$ as a function interatomic distance $r$(Å) of the semiconducting (8,0) zigzag SWCNTs under compressive -$\gamma$(%) and tensile +$\gamma$(%) strains at (a) $T$ = 100K, (b) $T$ = 400K, (c) $T$ = 700K and (d) $T$ = 1000K, respectively. The inset figures show the clear heights of 1$^{st}$ and 2$^{nd}$ peaks of $g(r)$ for varying ±$\gamma$(%) strains.

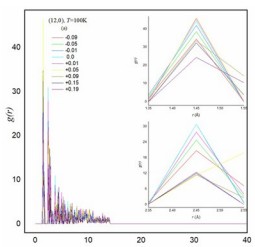 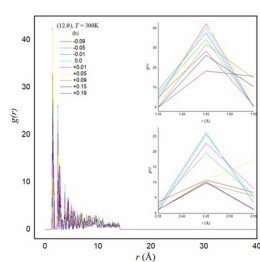 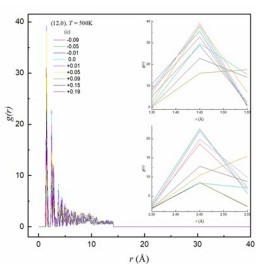 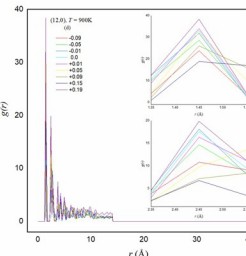

**Fig 5.** Structural analysis test of RDF $g(r)$ as a function interatomic distance $r(\text{Å})$ of the metallic (12,0) zigzag SWCNTs under compressive $-\gamma(\%)$ and tensile $+\gamma(\%)$ strains at (a) $T = 100K$, (b) $T = 300K$, (c) $T = 500K$ and (d) $T = 900K$, respectively. The inset figures show the clear heights of 1st and 2nd peaks of $g(r)$ for varying $\pm\gamma(\%)$ strains.

(100K). It is noted that the height of the peaks with tensile strain is definitely lower as compared to peaks height with compressive strain, especially for the range of $5 \leq +\gamma(\%) \leq 19$, however, the maximum peak height is observed at $+\gamma = 0\%$ to 1%. We can provide one of the possible reasons for reduction of peak heights that may cause stretching of the lattice structure of CNTs due to applied tensile strains, and this stretching is more pronounced due to its rigidness at low $T(K)$. In general, as tensile strain is applied to the CNTs, the distance between atoms $r(\text{Å})$ increases, leading to a decrease in the peak height of the RDF pattern. However, at a certain critical and or particular strain, the CNTs structure may become unstable and buckling process is started or breakdown of CNTs, which can result in a sudden increase/ decrease in the peak height. But, at intermediate-high $T$ (= 400K and 700K), the thermal fluctuations can feeble the CNTs structure [19] and make it more susceptible to break at low level [32] of $+\gamma$ strains. It can lead to abrupt and sudden changes (increase and or decrease) in the peak heights because of the CNTs structure breakdown at $T = 400K$ to 700K, and it can be seen in panels of Fig 4B and 4C. At high $T = 1000K$, in the last panel of Fig 4D, a decreasing pattern is observed in RDF peak heights for tensile strain $+\gamma$ (= 1% to 10%) range and it is the same pattern as observed at low $T = 100K$. As mentioned earlier that the high $T(K)$ may weaken the CNTs structure and it makes more inclined to break down the structure even at low $+\gamma$ strains. However, at tensile strains $+\gamma$ (= 15–19%) an abrupt increase in peak height is noted, indicating the CNTs structure breakdown and it can be seen in forthcoming Fig 6 (top portion). At higher $T(K)$, the thermal vibrations of atoms direct to a high average bond length [32] but the influence of stretching is relatively less.

In case of compressive strains, the pattern of RDF heights in metallic zigzag SWCNTs is same as in observed in semiconducting zigzag SWCNTs. It is examined that the peak heights of metallic SWCNTs are higher as observed in semiconducting SWCNTs, however, the peak heights (1st and 2nd) decrease with increasing $T(K)$. In the case of applied tensile strains at low $T$ (= 100K), the first peak [at $r(\text{Å}) = 1.45$] of RDF increases initially due to the elongation of the CNTs structure for $+\gamma$ (= 0 to 5%) but it then starts to decrease significantly because of the reduction in the nearest-neighbor distances between the carbon atoms for $9 \leq +\gamma(\%) \leq 19$. The second peak [at $r(\text{Å}) = 2.45$] initially decreases due to the reduction in the coordination number of the carbon atoms, but then starts to increase very slightly as the CNTs structure becomes more flexible without buckling. This pattern is observed for (12,0) zigzag SWCNTs under $+\gamma$ at $T = 100K$ and 300K, as shown in Fig 5A and 5B. For the scenario of high $T$ (= 900K), as illustrated in the last panel of Fig 4D, the pattern of RDF peaks under tensile strains indicates that the first peak [at $r(\text{Å}) = 1.45$] increases slightly at $+\gamma$ (= 1%) but it then decreases with an increase in $+\gamma$ (= 5% to 15%) due to the reduction in the coordination number of the carbon atoms. The CNTs structure then buckles at a certain strain $+\gamma$ (= 19% at $T = 900K$), which causes an increase in significantly height of the first and second peaks. This certain

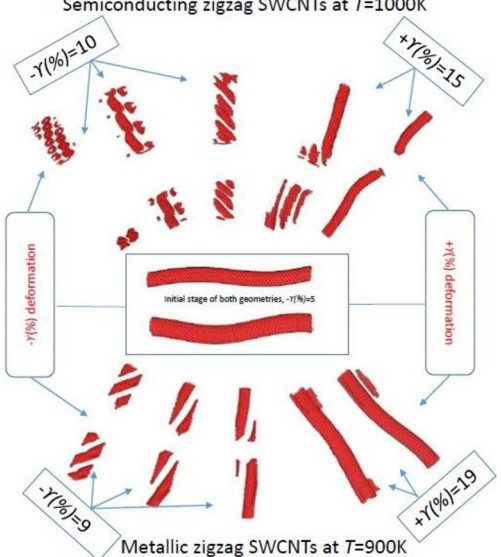

**Fig 6.** Visualization of the initial and final states of the zigzag SWCNTs regarding buckling and deformation processes of (top portion) semiconducting (8,0) SWCNTs at $T$ = 1000K and (bottom portion) metallic (12,0) zigzag SWCNTs $T$ = 900K under certain strains.

tensile $+\gamma$ (= 19%) value at which the structure of SWCNTs breakdown, resulting in an abrupt increase in the RDF peak heights, and it may be observed in forthcoming Fig 6 (bottom portion). It is concluded that the overall compressive strain reduces the peak heights while tensile strains initially increase peak heights for low $T$(K), however, it can lead to structural instability and abrupt increases in peak heights for high $T$(K). Consequently, the behavior of RDF peaks under varying strains is influenced by different factors such as temperature, coordination number, atom vibrations, and structural buckling or breaking.

Fig 6 illustrates the visualization of initial and final positions of the structures of semiconducting (top portion) and metallic (bottom portion) zigzag SWCNTs are captured as buckling (initial) and deformation (final) frames under different $\pm\gamma$(%) at $T$ = 1000K and 900K, respectively. This figure provides a visual representation of how $T$(K) and $\pm\gamma$(%) affect the structural properties of both zigzag SWCNTs, showcasing the changes in atomic positions and overall deformations in response to these conditions. Here, from the centre of Fig 6, the snapshots of CNT structures show the initial buckling of both zigzag SWCNTs with $-\gamma$ = 5%. To demonstrate the existence of final deformation of both zigzag SWCNTs, three snapshots are captured starting left (top and bottom) side at $-\gamma$(%) = 10, $-\gamma$(%) = 9 and two snapshots are recorded from the right (top and bottom) side at $+\gamma$(%) = 15, $+\gamma$(%) = 19, respectively, for the top (8,0) and bottom (12,0) SWCNTs. Fig 5 indicate the transformation process of zigzag SWCNTs concerning buckling and then reaches to deformation under varying $\pm\gamma$(%) strains, validating the previous armchair SWCNTs and single grapheme measurements of Li *et al.*, [2] and torsional deformation observations of Zhigilei *et al.*, [15]. For both zigzag SWCNTs, opposite behaviors are noted that the (8,0) SWCNT is able to bear more compressive strains than tensile strains while the (12,0) SWCNT demonstrates the opposite behavior as it tolerates more tensile strains than compressive strains. It is obvious that the (8,0) SWCNTs can bear less tensile ($-\gamma$ (%) = 10) and compressive ($+\gamma$(%) = 15) at $T$ (= 1000K) as compared to (12,0) SWCNTs where tensile ($-\gamma$(%) = 9) and compressive ($+\gamma$(%) = 19) strains are noted at $T$ (= 900K). It seems that earlier buckling and deformation processes are observed for (8,0) SWCNTs as contrast to

(12,0) SWCNTs, as already discussed and evident in our earlier Figs 4D and 5D. In the case of the (8,0) zigzag SWCNTs, the carbon atoms are arranged in a more compact arrangement, leading to stronger interatomic bonds. This makes the nanotube more resistant to$-\gamma$(%) strains, as the atoms can resist being pushed closer together. However, when the nanotube is under $+\gamma$(%) strains, the bonds among atoms are stretched and eventually break, leading to a decrease in the nanotube's strength [33]. On the other hand, in the case of the (12,0) zigzag SWCNTs, the atoms are arranged in a more open structure with weaker interatomic bonds. This makes the nanotube more resistant to tensile strains, as the atoms can stretch further before the bonds break. But, the nanotube is less resistant to compressive strains, as the weaker bonds among atoms can be easily pushed closer together, leading to buckling and ultimately breakdown the nanotube. Secondly, the distortions are more prominent for (8,0) SWCNTs because of its small diameter. Overall, the buckling and deformation behaviors of the (8,0) and (12,0) zigzag SWCNTs under $\pm\gamma$(%) strains are the consequence of the interplay between their crystal structure and the strength of the interatomic bonds.

## 3.2 Effects of strains and thermal conductivity

This part focuses on the thermal behavior of both zigzag SWCNTs when are subjected to a varying $\pm\gamma$(%) strains for a wide range of $T$(K). The knowledge of how strain affects thermal conductivity offers valuable insights for optimizing the design of nanoscale devices and materials to enhance thermal management capabilities. By understanding the impact of strains on thermal conductivity, researchers can develop strategies to improve the efficiency and performance of such systems in thermal management applications [34].

The behavior of thermal conductivity $\lambda(T)$ in accordance with $T$(K), when different $\pm\gamma$(%) strains are applied on (8,0) zigzag SWCNTs are displayed in Fig 7. At low $T$ (= 100K to 300K), the $\lambda(T, \gamma)$ of (8,0) zigzag SWCNTs gradually increases with an increase in $\pm\gamma$(%) strains and $T$ (K) meanwhile the buckling process is not observed, as shown in first panel (a). It is illustrated that the reported (8,0) SWCNTs may control under compressive strains of $0\leq -\gamma$(%) $\leq 10$ and tensile strains of $0\leq +\gamma$(%) $\leq 19$. For intermediate range $400\leq T$(K) $\leq 700$, as in Fig 7B, a converse effect is noted that the $\lambda(T, \gamma)$ decreases definitely with increasing $-\gamma$(%) and $T$(K), however, the $\lambda(T, \gamma)$ increases for the tensile strain range of $0\leq +\gamma$(%) $\leq 10$, same as observed for low $T$(K). It should be mentioned here that a buckling process is observed at $+\gamma$(%) = 15 and it reduces the $\lambda(T, \gamma)$ slightly for intermediate $T$(K) range. It is noted that the (8,0) SWCNTs experience fracture when subjected to a $+\gamma$(%) = 19 strain and it is concluded that (8,0) nanotube can withstand between $0\leq -\gamma$(%) $\leq 10$ and $0\leq +\gamma$(%) $\leq 15$ strains at intermediate $400\leq T$ (K) $\leq 700$ range. As the $T$(K) increases, the thermal vibrations of the atoms increase that lead to a decrease in the $\lambda(T, \gamma)$ [35]. In intermediate $T$(K) range, the semiconducting zigzag SWCNTs experience a significant reduction in $\lambda(T, \gamma)$ due to buckling under $\pm\gamma$(%) strains. Buckling causes a reduction in the $\lambda(T, \gamma)$ as the phonon scattering at the buckling points creates more resistance to the flow of heat [19]. Fig 7C depicts how thermal behavior is totally different at high $T$(K). At the high ($800\leq T$(K) $\leq 1000$) range, the (8,0) SWCNTs go under buckling process at earlier strains of $-\gamma$(%) = 5 and $+\gamma$(%) = 10, and soon after buckling a breakdown is started in (8,0) zigzag SWCNTs. This earlier buckling may decrease the $\lambda(T, \gamma)$ for the case of $-\gamma$(%) strains and increase for the case of $+\gamma$(%) strains and this buckling is due to increase in thermal vibrations of the atoms in the high $T$(K) range, making the (8,0) SWCNTs more risky to earlier deform and breakage [32]. But it is observed that the $\lambda(T, \gamma)$ definitely decreases as the $T$(K) increases (= 800K to 1000K) for $\pm\gamma$(%) strains, which is the same pattern of $\lambda(T, \gamma)$ as observed in intermediate $T$ (= 400K to 700K). From panels of Fig 7A–7C, it is demonstrated that the traceable domain of strains $\pm\gamma$(%) reduces from (-10%$\leq \gamma$

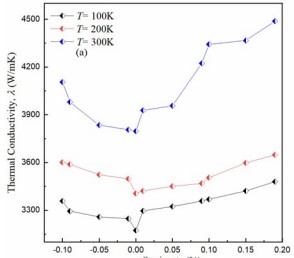 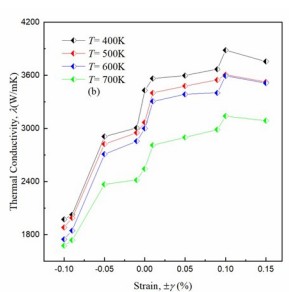 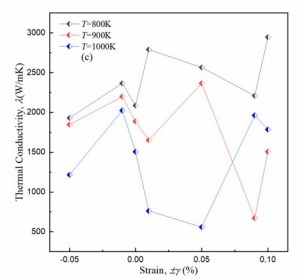 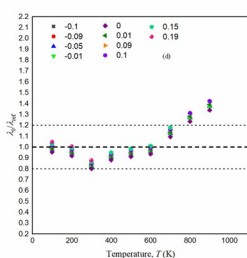

**Fig 7.** Variations of computed thermal conductivity $\lambda(T, \pm\gamma)$ of semiconducting (8,0) zigzag SWCNTs as a function of $\pm\gamma(\%)$ strains for (a) $T$ = 100K, 200K and 300K, (b) $T$ = 400K, 500K, 600K and 700K, (c) $T$ = 800K, 900K and 1000K, and (d) Comparison of calculated data $\lambda/\lambda_{REF}(T)$ of the present EMD method at different $T$(K) for the tunable regime of $\pm\gamma(\%)$. The current results obtained in panels (a), (b) and (c) at $+\gamma(\%)$ = 5 is taken as reference data. The small dotted lines show $\pm20\%$ deviation from the reference data (dotted line).

$\leq +19\%$) to (-5% $\leq \gamma \leq +10\%$) with increasing system $T$ (= 100K to 1000K), for both $\pm\gamma(\%)$ strains.

Now dealing with the (12,0) SWCNTs, as shown in Fig 8, a bit different trend is observed for $\lambda(T)$ under varying $\pm\gamma(\%)$ strains. It can be seen from Fig 8A that the $\lambda(T, \gamma)$ exhibits an increasing trend for both $\pm\gamma(\%)$ strains at low $T$ (= 100K to 400K) range. However, a fast increasing behavior of $\lambda(T, \gamma)$ is noted under tensile strains as compared to compressive strains, and $\lambda(T, \gamma)$ increases significantly with increasing system $T$(K). It can be attributed to the absence of buckling, allowing for efficient heat transfer through the (12,0) SWCNTs. It is evident that the (12,0) SWCNTs remain intact with an increase in strains in low $T$(K) range, enabling better $\lambda(T, \gamma)$ management. It is shown that the metallic zigzag SWCNTs can work efficiently for $0\leq -\gamma(\%) \leq 5$ and $0\leq +\gamma(\%) \leq 19$ at low $T$(K) range. As the $T$(K) is further raised to intermediate range, $500\leq T$(K) $\leq 600$, a different behavior is found in (12,0) SWCNTs that the $\lambda(T, \gamma)$ continues to rise steadily under $-1\leq \gamma(\%) \leq +15$ strain range, as displayed in Fig 8B. This increasing trend is same as observed for low $T$(K) range but with reduced $\pm\gamma(\%)$ range. However, beyond this $\pm\gamma(\%)$ strain range, a buckling process (at $-\gamma(\%)$ = 5 and $+\gamma(\%)$ = 19) is started that leads to a decrease in $\lambda(T, \gamma)$ sharply and this buckling introduces irregularities and disruptions in the nanostructure, hindering the heat transfer [15]. This decrease in $\lambda(T, \gamma)$ may be attributed to the excessive elongation and distortion of the (12,0) SWCNTs, causing structural damage and a disruption in heat conduction pathways [19]. It is concluded that the metallic zigzag SWCNTs may tune between an appropriate range of $-1\leq \gamma(\%) \leq +15$ strains for efficient work without breakdown of nanostructure. The third panel of Fig 8C shows that

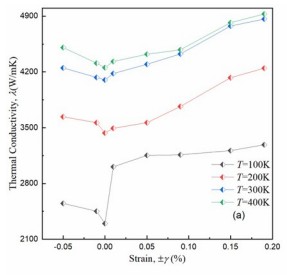 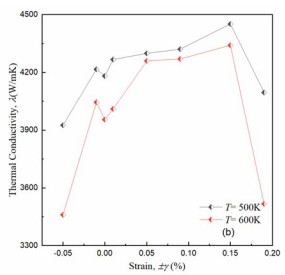 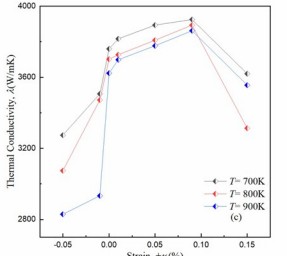 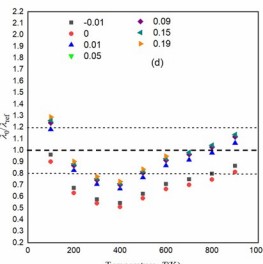

**Fig 8.** Variations of computed thermal conductivity $\lambda(T, \pm\gamma)$ of metallic (12,0) zigzag SWCNTs as a function of $\pm\gamma(\%)$ strains for (a) $T$ = 100K, 200K, 300K and 400K, (b) $T$ = 500K, and 600K, (c) $T$ = 700K, 800K and 900K, and (d) Comparison of calculated data $\lambda/\lambda_{REF}(T)$ of the present EMD method at different $T$(K) for the tunable regime of $\pm\gamma(\%)$. The current results obtained in panels (a), (b) and (c) at $-\gamma(\%)$ = 5 is taken as reference data. The small dotted lines show $\pm20\%$ deviation from the reference data (dotted line).

the $\lambda(T, \gamma)$ increases gradually for $0 \leq +\gamma(\%) \leq 8$ and it suddenly drops due to buckling process is observed at $+\gamma(\%) = 15$, for high $700 \leq T(K) \leq 900$. However, the $\lambda(T, \gamma)$ is definitely dropped for $0 \leq -\gamma(\%) \leq 5$ because of buckling at $-\gamma(\%) = 5$ and metallic SWCNT is fractured at $-\gamma(\%) > 5$ and $+\gamma(\%) > 15$. As mentioned above, this buckling causes the nanostructure to deform and lose its structural integrity, resulting in reduced heat conduction capacity. For both intermediate and high $T(K)$ cases, the $\lambda(T, \gamma)$ of (12, 0) zigzag SWCNTs decreases with increasing $T(K)$ same as observed for (8,0) zigzag SWCNTs. It can be interpreted from panels of Fig 8A–8C that the buckling process shifts to lower $\pm\gamma(\%)$ with increasing system $T$ (= 100K to 900K).

Panel (d) of Figs 7 and 8 compares the $\lambda(T)$ for different $\pm\gamma(\%)$ strains $\pm\gamma(\%)$, normalized by the reference data, calculated here from EMD of (8,0) and (12,0) zigzag SWCNTs, respectively, for varying strains with a reference set of data which is acceptable for nearly all $T(K)$. The reference set of data ($\lambda_{\text{ref}}$) is chosen as the outcomes from the EMD simulation, respectively, for (8,0) and (12,0) zigzag SWCNTs at $\gamma(\%) = +5$ and $\gamma(\%) = -5$ and is presented in panels (a), (b) and (c) of the respective Figs. Panel (d) indicates that the traceable regime is found approximately between $-10\% \leq \gamma \leq +19\%$ for both semiconducting and metallic zigzag SWCNTs, which depends on the nanotube $T(K)$. The reported $\pm\gamma(\%)$ points higher than the strained points display in panels (d) are also checked for various combinations of $T(K)$ and $L$ (Á) in order to establish the most suitable regime for the $\lambda(T)$ of semiconducting and metallic zigzag SWCNTs. The compressive and tensile strains of $-\gamma(\%) > 10$ and $+\gamma(\%) > 19$ provide more bucking or deformation measurements particularly for the higher $T(K)$. It is observed that our own data, normalized by the reference set of data, computed from the different varying $\pm\gamma(\%)$ points also included in panels (d) for comparison that are more close to each other and unstrained $\lambda(T)$ values. It is observed that the discrepancies among the own simulation results of (8,0) SWCNTs are large at high $T$ (= 800K, 900K) and the differences among the most of present data sets are, however, much smaller. These most of data points fall within less than a $\pm15\%$, however, at high $T(K)$, the deviation of the current data from the reference set of data is still reasonable and the data fall within between +20% to +30% range around the reference data. The obtained data of (12,0) SWCNTs lie below -30% range around reference line for $200 \leq T(K) \leq 700$ at few $\pm\gamma(\%)$ values, however, most of present data fall within $\pm20\%$ $w r t$ reference set of line. Comparison with own data in panels (d) of Figs 7 and 8 suggest that the measured $\lambda(T)$ is generally under predicting to within 1% - 20% relative to the own obtained data and most of current reported data points fall within the $\pm20\%$ range around the reference data, regardless of the $\pm\gamma(\%)$ strains in the EMD technique.

Fig 9 represents the variation of obtained results of (8,0) and (12,0) SWCNTs employing EMD simulation for a wide range of $100 \leq T(K) \leq 1000$. Our calculations for $\lambda(T)$ with semiconducting and metallic SWCNTs without strains as a function of system $T(K)$ are compared with the earlier results from MD simulations of Osman and Srivastava [8], EMD computations of Li *et al.*, [36], and Khan *et al.*, [37], GK homogeneous nonequilibrium MD (GK-HNEMD) of Zhang *et al.*, [38], EMD and NEMD estimations of Berber *et al.*, [27], and experimental investigations of Pop *et al.*, [9] and Yoshino *et al.*, [11]. The reported outcomes are in satisfactory agreement with earlier numerical and experimental investigations and illustrate that the current EMD simulations and previous GK-HNEMD and NEMD techniques have comparable efficiency, all producing nearly close values of $\lambda(T)$. The simulation results indicate that the (12,0) SWCNTs exhibit higher $\lambda(T)$ for intermediate to high $300 \leq T(K) \leq 900$, however, the (8,0) SWCNTs have definitely high $\lambda(T)$ for low $T$ (= 100K). This behavior can be attributed to the large diameter of the (12,0) zigzag SWCNTs, which results in low $\lambda(T)$ at low $T$ (= 100K) while increasing in $T(K)$ increases surface area and it facilities high heat transfer. Conversely, the smaller diameter of the (8,0) zigzag SWCNT may restrict heat transfer due to phonon

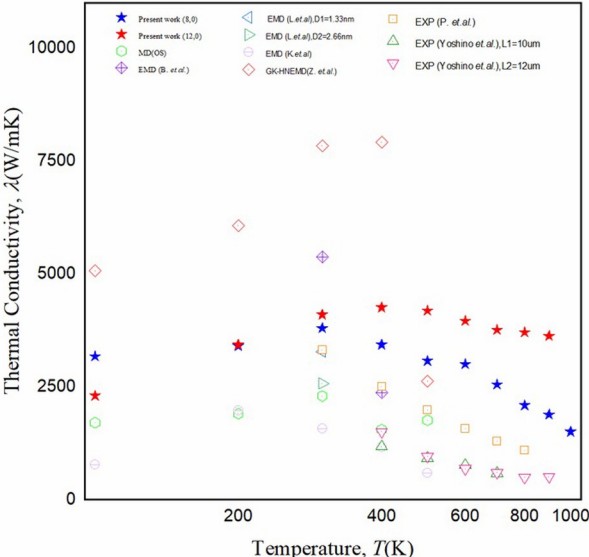

**Fig 9. Variations of obtained thermal conductivity λ (T) versus temperature T(K) for semiconducting (8,0) and metallic (12,0) zigzag SWCNTs with MD simulations of Osman and Srivastava [8]: MD (OS), EMD computations of Li *et al.*, [35]: EMD (L. *et.al.*) and Khan *et al.*, [36]: EMD (K. *et.al.*), GK homogeneous nonequilibrium MD of Zhang *et al.*, [37]: GK-HNEMD(Z. *et.al.*), EMD and NEMD estimations of Berber *et al.*, [27]: EMD (B. *et.al.*), NEMD (B. *et.al.*) and experimental investigations of Pop *et al.*, [9]: EXP (P. *et.al.*) and Yoshino *et al.*, [11]: EXP (Y.*et.al.*).**

confinement, leading to lower $\lambda(T)$ as compared to the (12,0) SWCNTs. These findings align with previous numerical and experimental results [9, 11] further supporting the validity of the present results. It is noteworthy that the $\lambda(T)$ of the (8,0) SWCNTs closely matches with the previous findings at all $T$(K) and this agreement validates the accuracy and consistency of the present results. Moreover, it should be mentioned here that the (12,0) zigzag SWCNTs break at $T$ (= 1000K), while the (8,0) zigzag SWCNTs may not withstand and it breaks at high at $T$ (= 900K). The breakdown behavior in the (12,0) zigzag SWCNTs can contribute the high thermal expansion coefficient of the larger diameter of CNTs, which causes larger thermal strains to develop at high $T$(K), leading to the breaking or buckling of the CNTs [38]. The diameter difference between the two SWCNTs and its impact on heat transfer may provide a plausible explanation for the observed trends.

## 4. Conclusions

We have computed the thermal conductivity $\lambda(T, \gamma)$ and structural analysis $g(r, \gamma)$ of (8,0) and (12,0) zigzag SWCNTs cross a range of system temperatures $T$ (= 100K, 1000K) with a constant nanotube length $L(\text{Å})$ and varying $\pm\gamma$(%)strains. The EMD simulations indicate that the present algorithm provides accurate analyses of $g(r, \gamma)$ and $(T, \gamma)$ with fast convergence and the metallic zigzag SWCNTs are more stable and resistive to changes as compared to semiconducting zigzag SWCNTs for an elevated $T$(K) range. The structural strength and stiffness of the zigzag SWCNTs have been further evaluated by varying compressive $-\gamma$ (= 1–10%) and tensile $+\gamma$ (= 1–19%) strains. The structural analysis indicates that the peak height reduces with an increase in $T$(K) at a certain $\pm\gamma$(%) where the both zigzag SWCNTs buckle and or break are observed, resulting in an abrupt increase in RDF peaks. The compressive and tensile buckling (and or deformation) of both zigzag SWCNTs has exposed that the deformation processes of

semiconducting configuration are more pronounced than that of metallic configuration. Simulation results show that the $\lambda(T)$ of metallic zigzag SWCNTs is higher as compared to semiconducting zigzag SWCNTs for $T \geq 200K$. The present EMD estimations predict that the structural and thermal analyses of (8,0) and (12,0) zigzag SWCNTs are helpful and essential for the understanding and development of high performance nanocomposites. It is shown that the obtained $\lambda(T, \pm\gamma)$ of (12,0) SWCNTs increases, however, the $\lambda(T, \pm\gamma)$ of (8,0) SWCNTs decreases with increasing $T$ (= 100K to 400K) and $T$ (= 400K to 1000K), respectively. The overall increasing trend of $\lambda(T, \pm\gamma)$ is noted for (8,0) and (12,0) SWCNTs with increasing $\pm\gamma$ at low $T$(K) range, however, tunable domain of strains $\pm\gamma$(%) reduces with an increase in $T$(K) and the corresponding changes in RDF peak heights, confirming our $\lambda(T, \pm\gamma)$ findings. The EMD estimations under varying $\pm\gamma$ strains are in satisfactory well matched with earlier numerical data of GK-HNEMD, NEMD, EMD and own reference data points and with experimental findings, and it displayed that the deviations are within less than ±20% for our data. For future work, it will be highlighted the importance of considering the size and chirality of SWCNTs when analyzing their thermal properties under varying strains. These outcomes will be pivotal in advancing our comprehension of the thermal transport mechanism in both zigzag SWCNTs subjected to mechanical deformation. Moreover, these insights can be instrumental in guiding the development and enhancement of nanoscale devices and materials.

## Supporting information

**S1 File.**
(TXT)

## Acknowledgments

We are very obliged to the National Advanced Computing Centre of National Centre for Physics (NCP), Pakistan for allocating computer time to test and run our MD code.

## Author Contributions

**Conceptualization:** Aamir Shahzad.

**Data curation:** Ama tul Zahra.

**Formal analysis:** Alina Manzoor, Jamoliddin Razzokov, Qurat ul Ain Asif, Kun Luo, Guogang Ren.

**Funding acquisition:** Kun Luo, Guogang Ren.

**Investigation:** Ama tul Zahra, Jamoliddin Razzokov, Qurat ul Ain Asif.

**Methodology:** Ama tul Zahra, Aamir Shahzad, Guogang Ren.

**Resources:** Aamir Shahzad.

**Software:** Ama tul Zahra.

**Supervision:** Aamir Shahzad.

**Validation:** Alina Manzoor, Jamoliddin Razzokov, Kun Luo.

**Visualization:** Qurat ul Ain Asif.

**Writing – original draft:** Ama tul Zahra, Aamir Shahzad.

**Writing – review & editing:** Alina Manzoor, Jamoliddin Razzokov, Kun Luo, Guogang Ren.

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
