## [Decision Letter · Decision Letter 0]

21 Dec 2023

Structural and Thermal Analyses in Semiconducting and Metallic Zigzag Single-Walled Carbon Nanotubes using Molecular Dynamics Simulations

PONE-D-23-35788

Dear Dr. Shahzad,

We’re pleased to inform you that your manuscript has been judged scientifically suitable for publication and will be formally accepted for publication once it meets all outstanding technical requirements.

Kind regards,

Anshu Sharma, Ph.D.

Academic Editor

PLOS ONE

Journal Requirements:

"National Natural Science Foundation of China (No. 51874051, 52111530139) and award from the UK Royal Society (IEC\\NSFC\\201155, 2021-23)"

Please respond by return e-mail so that we can amend your financial disclosure and competing interests on your behalf.

4. Please note that PLOS ONE has specific guidelines on code sharing for submissions in which author-generated code underpins the findings in the manuscript. In these cases, all author-generated code must be made available without restrictions upon publication of the work. Please review our guidelines at https://journals.plos.org/plosone/s/materials-and-software-sharing#loc-sharing-code and ensure that your code is shared in a way that follows best practice and facilitates reproducibility and reuse.

Additional Editor Comments (optional):

Manuscript entitled Structural and Thermal Analyses in Semiconducting and Metallic Zigzag Single-Walled Carbon Nanotubes using Molecular Dynamics Simulations has been reviewed. Reviewers recommended for the accpetance of the manuscript.

Reviewers' comments:

Reviewer's Responses to Questions

**Comments to the Author**

1. Is the manuscript technically sound, and do the data support the conclusions?

Reviewer #1: Yes

Reviewer #2: Yes

2. Has the statistical analysis been performed appropriately and rigorously? 

Reviewer #1: Yes

Reviewer #2: Yes

3. Have the authors made all data underlying the findings in their manuscript fully available?

Reviewer #1: Yes

Reviewer #2: Yes

4. Is the manuscript presented in an intelligible fashion and written in standard English?

Reviewer #1: Yes

Reviewer #2: Yes

5. Review Comments to the Author

Reviewer #1: Zahra and co-author created a study dealing with structural and thermal analyses in semiconducting and metallic zig-zag single-walled carbon nanotubes using molecular dynamics simulations. In summary the presented manuscript represents an interesting study to which I have no reservations and for the reason mentioned I can only recommend acceptance.

Reviewer #2: The authors had analysed the Semiconducting and Metallic Zigzag Single-Walled Carbon Nanotubes using Molecular Dynamics Simulations very well. All the analysis and description of the related to CNTs are well interpreted.

6. PLOS authors have the option to publish the peer review history of their article (what does this mean?). If published, this will include your full peer review and any attached files.

Reviewer #1: No

Reviewer #2: **Yes: **Dr. Jaymin Ray

---

## [Editor Report · Acceptance letter]

1 Feb 2024

PONE-D-23-35788 

PLOS ONE

Dear Dr. Shahzad, 

I'm pleased to inform you that your manuscript has been deemed suitable for publication in PLOS ONE. Congratulations! Your manuscript is now being handed over to our production team.

Kind regards, 

on behalf of

Dr. Anshu Sharma 

Academic Editor

PLOS ONE